# Role of Insulin-like Growth Factor-1 Receptor in Tobacco Smoking-Associated Lung Cancer Development

**DOI:** 10.3390/biomedicines12030563

**Published:** 2024-03-02

**Authors:** Ayaz Shahid, Shaira Gail Santos, Carol Lin, Ying Huang

**Affiliations:** 1Department of Pharmaceutical Sciences, College of Pharmacy, Western University of Health Sciences, Pomona, CA 91766, USA; carol.lin@westernu.edu; 2College of Osteopathic Medicine of the Pacific, Western University of Health Sciences, Pomona, CA 91766, USA

**Keywords:** Insulin-like Growth Factor 1 Receptor, lung cancer, tobacco smoke carcinogens

## Abstract

Cancer remains a significant global health concern, with lung cancer consistently leading as one of the most common malignancies. Genetic aberrations involving receptor tyrosine kinases (RTKs) are known to be associated with cancer initiation and development, but RTK involvement in smoking-associated lung cancer cases is not well understood. The Insulin-like Growth Factor 1 Receptor (IGF-1R) is a receptor that plays a critical role in lung cancer development. Its signaling pathway affects the growth and survival of cancer cells, and high expression is linked to poor prognosis and resistance to treatment. Several reports have shown that by activating IGF-1R, tobacco smoke-related carcinogens promote lung cancer and chemotherapy resistance. However, the relationship between IGF-1R and cancer is complex and can vary depending on the type of cancer. Ongoing investigations are focused on developing therapeutic strategies to target IGF-1R and overcome chemotherapy resistance. Overall, this review explores the intricate connections between tobacco smoke-specific carcinogens and the IGF-1R pathway in lung carcinogenesis. This review further highlights the challenges in using IGF-1R inhibitors as targeted therapy for lung cancer due to structural similarities with insulin receptors. Overcoming these obstacles may require a comprehensive approach combining IGF-1R inhibition with other selective agents for successful cancer treatment.

## 1. Introduction

Cancer is among the leading causes of death worldwide and was responsible for nearly 10 million deaths in 2020 [1]. Among different types of cancer, lung cancer is the most common type of cancer globally [2], causing the most cancer-related deaths, with approximately 1.8 million deaths globally (18% of cancer-related deaths) [2,3]. In recent years, lung cancer has seen a rapid increase in both prevalence and mortality in industrially developed countries [4]. 

Receptor tyrosine kinases (RTKs) are a group of cell membrane receptors that play a vital role in multiple cellular processes, such as cell growth, differentiation, proliferation, and survival [5]. Many studies have shown that the dysregulation of RTK signaling can lead to various diseases, including cancer [6]. One of the members of the RTK family is the Insulin-like Growth Factor 1 Receptor (IGF-1R) [7], which plays a crucial role in mediating the biological effects of insulin-like growth factors (IGFs), specifically IGF-1 and partly IGF-2, which are produced in response to the release of growth hormone from the pituitary gland [7]. IGF-1R is also known for its oncogenic potential, which means that its abnormal activation or overexpression can contribute to the development and progression of various cancer types, including lung cancer [8,9,10,11]. Therefore, IGF-1R has been identified as a promising target for cancer treatment. The targeted inhibition of IGF-1R in cancer treatment yields more benefits than traditional non-selective treatments like chemotherapy and radiation therapy, which can harm normal cells [12]. However, IGF-1R inhibition can also lead to metabolic abnormalities, such as insulin dysregulation, through the action of IGF-1R/insulin receptor (IR) hybrids (as explained in Figure 1) [13,14]. Therefore, more investigations are needed in this area to determine the safest and most effective application of IGF/IGF-1R inhibitors as a therapy for lung cancer. Overall, the IGF/IGF-R signaling axis shows promise as a potential target for lung cancer therapies, but poses some challenges that cannot be overlooked.

Tobacco smoking is the leading cause of cancer-related deaths in both men and women in the United States [15]. Although extensive research has already been conducted, there are still unknown pathways that are activated by tobacco smoke-related carcinogens. Numerous studies are currently underway to address this research gap [16]. Targeting signaling pathways activated by tobacco smoke carcinogens is important in preventing lung cancer, as tobacco smoking causes one-third of all cancer-related deaths annually [17]. Tobacco smoke is comprised of thousands of chemicals, including at least 70 compounds known to cause cancer [18]. These cancer-causing agents can affect cell proliferation and promote uncontrolled cell growth by abnormally activating growth receptors [19,20]. Various studies have shown that tobacco smoke carcinogens can activate growth factor receptors, including IGF-1R, which play an important role in lung cancer development [21]. In this review article, we will discuss how tobacco smoke-specific carcinogens and IGF-1R signaling pathways are connected to lung cancer development, and how IGF-1R can serve as a potential target for lung cancer therapy.

## 2. What Is the IGF/IGF-R Signaling Axis?

The IGF/IGF-R signaling axis consists of two RTKs, IGF-1R and IGF-2R (Figure 1), along with their respective ligands, IGF-1/2, and binding proteins in the serum known as insulin-like growth factor binding proteins (IGFBPs) that act as transporter proteins [22,23]. The binding of IGFs to the IGF-Rs results in the subsequent downstream activation of several pathways that are crucial in cell proliferation and cell cycle progression [13] (Figure 2). Due to this mechanism, the IGF/IGF-R axis plays significant roles in many biological processes, such as normal cell growth and survival, and contributes to numerous pathological processes, including cancer development and progression [24]. 

The IGF/IGF-R axis is a highly conserved signaling pathway that has been around for millions of years, even before mammalian vertebrates existed [25,26]. While this pathway plays a crucial role in normal organismal growth and survival, it has also been shown to regulate animal carbohydrate and lipid metabolism [27,28,29,30,31,32]. The IGF-R is widespread in the body, and nearly every tissue can produce IGFs, making it an important research area due to its diverse biological effects and therapeutic potential [28,33,34]. It has also been demonstrated that the IGF/IGF-R axis is involved with numerous endocrine signaling pathways [28,35], which highlights the pleiotropic effects of IGF-R activation through the action of receptor hybrids composed of IGF-1R and IR monomers.

Similar to the IGF-R, IR is also a cell surface receptor that initiates downstream signaling cascades within the cell (Figure 1). However, both receptors respond to different ligands, as they differ slightly in two domains involved in ligand specificity: the first leucine-rich region, L1, and a cysteine-rich region, CR [34,36]. IR is mainly activated by insulin, whereas IGF-1R is activated by IGF-1 and IGF-2, leading to distinct signaling pathways upon activation of both receptors [37,38]. IR activation plays an important role in metabolic homeostasis by regulating glucose transport and glycogen synthesis [39,40]. Conversely, the activation of IGF-1R triggers PI3K and Ras-mediated pathways, primarily resulting in cell proliferation, growth, and survival [40,41] (Figure 2). The dysregulation of IGF-1R and IR signaling has been implicated in various diseases. For example, aberrant IR signaling is associated with type 1 and 2 diabetes [38]. As discussed earlier, abnormal IGF-1R activation has been associated with cancer initiation and development.

When exploring the IGF-1R signaling axis, it is imperative to consider the role of IR. Because of the structural homology between these two receptors, it has been found that IGF-1R monomers have the ability to dimerize with IR monomers, particularly the A isoform (IR-A), to create a functional hybrid receptor that can bind both IGFs and insulin [40] (Figure 1). IR-A is predominantly found in fetal and cancer cells and plays a crucial role in promoting cell cycle progression [34]. Ongoing research is being conducted to understand the roles of hybrid IGF-1R/IR receptors in various diseases and cellular processes, which is crucial for unraveling the complexities underlying cancer initiation and progression, as well as informing the development of targeted therapies. 

## 3. Structure and Functions of IGF-1R and its Components

The IGF-1R is a crucial component of the IGF/IGF-R signaling system, composed of two extracellular α subunits (IGF-1Rα) and two transmembrane β subunits (IGF-1Rβ) [22] that are produced by a single gene located on chromosome 15q26.3 [42]. The IGF-1Rα and IGF-1Rβ subunits form a tetramer [33] that plays a vital role in cell survival and transformation due to its tyrosine kinase activity [23]. IGF-2R, also known as the mannose 6-phosphate receptor, does not possess tyrosine kinase or signal transduction activity. Instead, IGF-2R helps regulate IGF-2 levels in the body by capturing and transporting it to the lysosome for degradation [23,43,44] (Figure 1). IGF-1 and IGF-2 can bind to IGF-1R; however, IGF-1R has a 15- to 20-fold higher affinity for IGF-1 than IGF-2 [45]. The half-life and bioavailability of IGF-1 and IGF-2 in the circulation vary depending on the level of IGFBPs in the serum, which have different binding affinities and specificities [46,47]. A lack of IGF-1/2 ligands can hinder growth and lead to metabolic issues [48]. On the other hand, high levels of IGF-1/2 are linked to a higher risk of cancer and a lower lifespan [8,49]. 

When IGF-1 and IGF-2 bind to IGF-1R, they activate the receptor’s tyrosine kinase domain, phosphorylating receptor substrates. This activation leads to two main signaling pathways: (1) the PI3K pathway initiated by insulin receptor substrate (IRS), which mainly affects metabolic outcomes, and (2) the Ras/mitogen-activated protein kinase (MAPK) pathway initiated by Src homology and collagen (Shc) adaptor protein, which mainly affects cell growth and differentiation [13,50] (Figure 2). Thus, the activation of IGF-1R simultaneously promotes cell survival and proliferation while providing inhibitory protective mechanisms against apoptosis [51,52]. Studies have demonstrated that IGF-1R overexpression was associated with disease progression, poor prognosis, and treatment resistance in breast cancer, esophageal adenocarcinoma, colorectal cancer, and squamous cell carcinoma [53]. A meta-analysis has also shown that IGF-1R expression is an unfavorable factor for survival in non-small cell lung cancer (NSCLC) patients, and is associated with smoking status and tumor size [53].

The IGFBP family is evolutionarily ancient and highly conserved in vertebrates [54,55]. Six types of IGFBPs, designated IGFBP-1 through IGFBP-6, have been identified in the serum and serve to decrease the amount of free circulating IGFs through varying binding affinities and specificities [46]. The most critical binding protein, IGFBP-3, binds up to 80% of IGF-1 in the serum [23] with a higher affinity for IGF-1 than IGF-1R possesses [56]. Studies have found that high IGF-1 levels in the blood may increase the risk of lung cancer, while high levels of circulating IGFBP-3 may decrease risk [45,57,58]. Previous studies have demonstrated that p53 regulates the expression of IGFBP-3 and that increased transcription leads to higher expression levels [59,60]. However, when p53 is mutated, it cannot activate IGFBP-3, which thus fails to induce apoptosis [61,62,63]. While the expression of IGFBP-3 may not be completely absent, a decrease in its expression can cause an elevation in the local availability of IGF-1 in lung tissue, thereby increasing the risk of developing lung cancer [64,65]. These data suggest that IGFBP-3 could be a promising therapeutic option for cancer treatment. 

## 4. IGF-1R Overexpression and Activation in Lung Cancer Initiation and Development

IGF-1R, an RTK activated by IGF-1 and IGF-2, contributes to lung cancer by initiating pathways involved in cell proliferation and survival [26]. Several studies have shown that IGF-1R activation leads to the opening of voltage-gated calcium channels, leading to an increase in intracellular Ca^2+^ levels [66,67]. The increased Ca^2+^ increases the phosphorylation of cAMP response element-binding protein (CREB) via the Erk5 pathway, resulting in the proliferation and transformation of lung epithelial cells [67]. This Ca^2+^ signaling mechanism leading to uncontrolled cell growth is crucial in promoting the formation and expansion of lung tumors [68,69]. Another study showed a significant increase in the expression of IGF-1 and IGF-1R in high-grade dysplastic bronchial tissues compared to normal tissues [70]. Additionally, the same study discovered that preneoplastic human bronchial epithelial (HBE) cells produce autocrine IGFs, which have the potential to elicit spontaneous lung tumor formation, especially after exposure to tobacco smoke carcinogens [70,71]. Some studies have also shown that IGFs are produced at high levels by the stromal tissue surrounding cancer cells, creating a favorable microenvironment for tumorigenesis and providing a prognostic factor for outcomes in lung cancer [10,72]. From these findings, it can be concluded that high IGF expression and signaling is a vital component of early lung carcinogenesis, transforming cells exposed to tobacco smoke carcinogens. The increased activity occurs through the activation of biochemical signaling pathways and the induction of DNA mutations, both caused by tobacco carcinogens [70].

In addition, IGF-1R has been well-studied for use in prognostic predictions in various malignancies, such as breast, prostate, lung, and colorectal cancers [73,74,75,76,77,78,79]. Two meta-analyses showed that a high expression of IGF-1R was associated with poor disease-free survival in non-small cell lung cancer (NSCLC) [53,80], a malignancy that is predominantly linked to tobacco smoking. However, one of these analyses found that poor overall survival in patients with NSCLC and small cell lung cancer (SCLC) could not be predicted based on IGF-1R expression [80]. Similarly, another study could not find a conclusive association between the prognosis of patients with SCLC and IGF-1R expression [81]. These findings suggest that IGF-1R may participate in developing NSCLC and could serve as a predictive prognostic factor for this specific cancer type.

## 5. Implication of IGF-1R Signaling Pathways in Tobacco Smoke-Associated Carcinogenesis 

It is well-established that the overexpression of IGF-1R and IGFs, as well as the deregulation of IGF-1R signaling pathways, are closely associated with the development of lung cancer [70]. Given that tobacco smoking is the main risk factor for lung cancer, several studies have shown that tobacco smoke-specific carcinogens activate IGF-1R signaling pathways [82,83,84]. For example, a study reported that the NSCLC tumor tissues isolated from smokers showed significantly greater levels of phosphorylated IGF-1R (pIGF-1R) compared with tissues from non-smokers. They also found that nicotine-derived nitrosamine ketone, 4-(methylnitrosamino)-1-(3-pyridyl)-1-butanone (NNK), a carcinogen specific to tobacco, stimulates the rapid Ca^2+^-mediated exocytosis of IGF-2, which can then bind to and activate IGF-1R [84]. Additionally, NNK was also shown to directly induce increased levels of pIGF-1R in NSCLC cells via β-adrenergic (β-AR) signaling. Treatment with β-AR antagonists resulted in the suppression of NNK-mediated cell transformation in human bronchial epithelial (HBE) cells and tumor formation in mouse models [83]. 

Furthermore, tobacco smoke works in conjunction with IGFs by accelerating its activity as an autocrine ligand for IGF-1R activation in lung epithelial cells [70,84]. Tobacco smoking also leads to disruptions in the renin-angiotensin system, which has been implicated in lung cancers and has been shown to mediate the activation of the IGF-1R signaling axis [82]. As described previously, the activation of IGF-1R can elicit spontaneous lung tumor formation through the aberrant activation of pathways involved in cell growth and survival. Thus, these findings suggest that tobacco smoking promotes lung tumorigenesis and the progression of malignancy through dysregulated IGF-1R activation. Below, we discuss potential downstream events after IGF-1R activation that contribute to transforming cells exposed to tobacco smoke carcinogens. 

### 5.1. Overactivation of IGF-1R Increases Proliferation and Metastasis of Lung Cancer

It has been demonstrated that the expressions of IGF-1 and IGF-2 increase during lung carcinogenesis caused by tobacco smoking, and the expression and phosphorylation of IGF-1R are increased in high-grade dysplastic lesions compared to standard lung tissues [70]. Upon activation, IGF-1R is phosphorylated, resulting in a signaling cascade that stimulates downstream pathways involved in cell proliferation and cell cycle progression, including the P13K/Akt/mTOR and Ras/Raf/MAPK pathways [13]. The activation of IGF-1R has anti-apoptotic effects on cancer cells, which contributes to cancer development and progression [85,86]. For tumor cells to spread to distant organs, they must be able to detach from the extracellular matrix. IGF-1R signaling can prevent cancer cells from undergoing anoikis, a type of cell death that occurs when cells lose their attachment to the extracellular matrix and neighboring cells, possibly via inhibiting the activation of p53 and p21 [87]. The activation of IGF-1R also promotes anchorage-independent growth by recruiting RACK1-mediated STAT3, an important molecular mediator for cell growth in a non-adherent manner [82,88,89]. According to Yang et al., deficiency of IGF-1R in the lung tumor microenvironment resulted in decreased tumor initiation and progression, due to reduced proliferation, vascularization, fibrosis, and inflammation, as well as immunosuppression [90]. Therefore, it can be concluded that IGF-1R activation is crucial for the anchorage-independent growth of cancer cells, and it promotes tumor metastasis and the progression of malignancy [87]. 

### 5.2. Overactivation of IGF-1R Promotes Epithelial–Mesenchymal Transition (EMT) and Stemness of Cancer Cells 

EMT is a critical event in the cellular evasion of death signals and an essential hallmark of cancer metastasis [91,92]. The significant changes during the EMT include the downregulation of epithelial markers, the rearrangement of the cytoskeleton, the loss of cell–cell adhesion and apical–basal polarity, and the acquisition of mesenchymal phenotype markers [93,94]. These cellular changes collectively drive cancer cells into a stem cell-like state, imparting the unique ability of self-renewal that significantly increases the difficulty of treating cancer [95]. 

Various studies have demonstrated that IGF-1R signaling induces EMT through multiple mechanisms in different cancers, including breast, prostate, and lung [96,97,98,99,100]. One proposed mechanism by which this occurs is through the IGF-1R activation of the PI3K/Akt/mTOR pathway, which leads to the upregulation of transcription factors like ZEB1 [95] and the stabilization of SLUG [101,102]. Both ZEB1 and SLUG function as negative regulators of E-cadherin expression, contributing to forming a mesenchymal phenotype [98]. As an alternative mechanism, it has also been demonstrated that IGF-1R may trigger EMT via interactions with STAT3, FAK, and NF-kB [101,103,104]. Given these findings, it appears that IGF-1R induces EMT and stem cell-like properties in cancer cells through numerous downstream targets, thereby increasing the chances of tumor metastasis to other sites in the body. 

## 6. IGF-1R and Anti-Cancer Drug Resistance in Lung Cancer 

Chemotherapy is commonly used to treat cancer, but resistance to chemotherapeutic agents often leads to treatment failure [105]. Some common resistance mechanisms include changes in the target proteins to decrease binding affinity, enhanced efflux pump to remove the drug from the cytoplasm, the absence of cellular mechanisms that result in apoptosis, and the use of alternative activating signaling pathways [105,106]. Accumulated evidence suggests that the activation of the IGF-1R signaling pathway plays an important role in chemotherapy resistance. Previous studies have shown that the overexpression of IGF-1R is associated with poor chemotherapy outcomes, while the inhibition of IGF-1R leads to improved chemotherapy responses in different types of cancers [9]. IGF-1R-mediated pathways promote cellular proliferation, inhibit apoptosis, alter drug targets, and increase the expression of transporter proteins to reduce intracellular drug concentration, all of which contribute to chemotherapeutic resistance [107]. 

The role of the IGF-1R signaling pathway is crucial in developing treatment resistance in lung cancer [108]. A study conducted on NSCLC cells that had developed resistance to cisplatin showed a decrease in the expression of IGFBP-3 and an increase in the activation of IGF-1R signaling, leading to the development of cisplatin resistance in NSCLC [109]. The resistance was reversed when human recombinant IGFBP-3 was used or when IGF-1R was inhibited using siRNA [109]. In another study on cisplatin resistance in NSCLC cells, the loss of IGFBP-3 expression resulted in reduced tumor cell sensitivity to cisplatin [110]. These findings indicate that components of the IGF-1R signaling axis, including IGFBP-3, play important roles in the cellular development of drug resistance to commonly used lung cancer therapies. 

Furthermore, the IGF-1R signaling pathway is associated with resistance to epidermal growth factor receptor (EGFR) tyrosine kinase inhibitors (TKIs) used for the treatment of NSCLC. EGFR is overexpressed in a majority of NSCLC cases; however, two available EGFR TKIs, erlotinib and gefitinib, have yielded poor responses in NSCLC patients with a history of smoking compared to patients who have never smoked [111]. It was also shown that the prolonged use of erlotinib or gefitinib for NSCLC resulted in acquired drug resistance mediated by the upregulation of IGF-1R [112,113]. According to Hurbin et al., the inhibition of IGF-1R with anti-IGF-1R antibodies re-sensitized resistant cells to gefitinib and induced apoptosis [114]. Acquired resistance to afatinib, another available EGFR TKI given to NSCLC patients with known EGFR mutations, is also attributed to the activation of the IGF-1R signaling pathway. The knockdown of IGF-1R sensitized resistant cells to afatinib and induced apoptosis in those cells [115]. In addition, NSCLC cells often develop resistance to osimertinib, a newer generation EGFR TKI used as a first- or second-line treatment for lung cancer. This resistance is mediated by IGF-1R activation through IGF-2 autocrine signaling [116]. These studies suggest that IGF-1R inhibition or a combination of EGFR/IGF-1R inhibition strategies could be potential therapeutic options for lung cancer patients who develop resistance to targeted EGFR inhibitors. 

The inhibition of IGF-1R has also been revealed to improve therapeutic response to anti-cancer agents in cancers that do not involve EGFR mutations. In a study of a patient with anaplastic lymphoma receptor tyrosine kinase (ALK) fusion-positive lung cancer who had developed TKI resistance, combined ALK and IGF-1R inhibition improved therapeutic efficacy [117]. Another study also found that the overexpression of IGF-1R resulted in cells with acquired resistance to PI3K inhibitors [103]. The targeted knockdown of IGF-1R expression using specific siRNAs or IGF-1R TKIs reversed the resistance. These results suggest that long-term treatment with PI3K inhibitors may cause reversible drug resistance, and targeting IGF-1R provides a promising strategy to overcome the resistance [118]. Overall, the IGF-1R signaling pathway plays a crucial role in the cellular development of anti-cancer drug resistance, and the use of IGF-1R inhibitors in conjunction with other therapies offers a promising direction for lung cancer treatment.

## 7. The IGF/IGF-1R Signaling Axis Is a Potential Target for Cancer Therapy

The IGF-1R signaling pathway offers promising targets for cancer therapy. Using monoclonal antibodies, small molecule tyrosine kinase inhibitors, and IGF-1R mutations and silencing are four common anti-cancer strategies targeting the IGF signaling system. These strategies are used to treat various types of cancer, including lung cancer [11] (Figure 3). Since pre-clinical studies may contribute to the decision-making process for advancing these therapies to clinical trials, various pre-clinical studies have investigated IGF-1R-targeted therapies in lung cancer models, using monoclonal antibodies and tyrosine kinase inhibitors (Table 1). Monoclonal antibodies that target IGF-1R, such as cixutumumab, dalotuzumab, and figitumumab, are used for both in vitro and in vivo lung cancer treatments. Similarly, tyrosine kinase inhibitors like linsitinib, picropodophyllin, and BMS-754807 can also target IGF-1R for lung cancer therapy (Table 1). Clinical studies involving IGF-1R-targeted therapies in lung cancer patients have been conducted to evaluate the effectiveness of these therapies (Table 2). Presently, numerous clinical trials are underway to evaluate the effectiveness of targeting IGF-1R (Table 2). For example, a phase II clinical trial was conducted to investigate the effectiveness of figitumumab (CP-751,871), an IGF-1R monoclonal antibody, in combination with chemotherapy (carboplatin and paclitaxel) for treating advanced NSCLC in patients. Although pre-clinical studies demonstrated promising results, the results from clinical trials showed that the combination therapy did not significantly improve overall survival compared to chemotherapy alone. However, it did demonstrate potential benefits in specific patient subgroups, highlighting the complexity of patient selection for IGF-1R-targeted therapy [119]. 

Furthermore, another study found that cancer cells were sensitized to cisplatin upon IGF-1R inhibition, which can occur through IGF-1R inhibitors or siRNA-mediated IGF-1R suppression [151]. Combining IGF-1R inhibition with other selective inhibitors of PI3K or ATR, a kinase involved in the cellular response to DNA damage, displayed synergistic therapeutic effects [151,152]. Also, combining an IGF-1R monoclonal antibody and rapamycin led to complete tumor regression in pediatric sarcoma, surpassing the effects of single therapeutic agents that only induced modest growth delay [153]. Other pre-clinical studies utilizing IGF inhibitors have exhibited synergism when used with other chemotherapy agents or radiation therapy [23]. It has been found that some microRNAs (miRNAs) are likely to affect the expression of IGF-1R or its receptor signaling activity [154]. The findings of this study also indicate that PI3K and AKT, which are components downstream of the IR, play a crucial role in mediating signals for the IGF-1R pathway. Additionally, further investigations into miRNAs could provide valuable insights because of their indirect impact on the function of IGF-1R. Because insulin resistance involves the dysregulation of multiple growth factor signals, including IGF-1R, miRNAs affecting IR sensitivity are also likely to impact the IGF-1R pathway. These collective endeavors highlight the potential to use IGF-1R inhibition as a potent anti-cancer therapy, mainly when employed in conjunction with complementary treatment modalities. 

As discussed previously, the IGF-1R signaling axis plays a critical role in many cellular processes. Consequently, IGF-1R-targeted therapies for any component of the IGF-1R pathway exhibit anti-cancer effects, such as inhibiting cell proliferation, inducing apoptosis, reducing angiogenesis, and reversing drug resistance (Table 3). All these mechanisms work together to synergistically enhance the therapeutic effectiveness of IGF-1R-targeted therapies in cancer treatment.

Unfortunately, despite these promising pre-clinical and early-phase clinical findings, numerous large-scale clinical trials involving NSCLC have yielded disappointing results, potentially due to a lack of patient selection markers [14,22,163]. One study on IGF-1R expression in NSCLC found that treatment with IGF-1R TKIs exhibited significant anti-tumor activity in NSCLC cells with wild-type EGFR and KRAS. Anti-tumor efficacy was not seen in NSCLC cells with mutations in those genes, indicating that EGFR and KRAS mutation status can be predictive markers of response to IGF-1R TKIs [164]. Other early-phase clinical trials have provided evidence that high levels of circulating IGF-1 in NSCLC cases can serve as a potential selection marker for response to IGF-1R inhibition [22,165]. These findings underscore the importance of refining patient selection criteria to harness the potential of IGF-1R inhibition in lung cancer treatment. 

## 8. Challenges in Targeting the IGF/IGF-1R Signaling Axis

While the IGF/IGF-R signaling axis shows promise as a potential target for lung cancer therapies, previous studies have revealed that targeting this pathway presents challenges that cannot be overlooked. As discussed previously, the ability of IGF-1R monomers to dimerize with the IR-A isoform and bind both IGFs and insulin poses a considerable obstacle [14]. Given this, the targeted inhibition of IGF-1R can also lead to metabolic abnormalities, such as hyperglycemia and insulin dysregulation, through the action of IGF-1R/IR-A hybrids [13,14]. Thus, further investigations exploring the complex interplay between IGF-1R, and IR must be undertaken in order to understand the nuances associated with these signaling pathways. In addition, while many clinical trials have tested the efficacy of drugs targeting the IGF/IGF-1R signaling axis, little work has been done to elucidate specific patient biomarkers that can predict therapeutic response [163,166,167].

Consequently, previous clinical trials using monoclonal antibodies and tyrosine kinase inhibitors against IGF-1R have failed to demonstrate a significant benefit for cancer patients, likely due to a lack of patient selection [151,162,168]. Because many successful targeted anti-cancer therapies have relied heavily on the use of predictive biomarkers [166], more investigation into this area is crucial for determining the most effective application of IGF/IGF-1R inhibitors as a lung cancer therapy. These challenges emphasize that a simplistic approach will not suffice, and that drug-based inhibition of any component of the IGF-1R signaling system should be applied in conjunction with chemotherapy agents. Further exploration in this research field is required, primarily focusing on circumventing IGF-1R/IR-A hybrids and identifying specific patient biomarkers. However, despite these important considerations, the IGF/IGF-1R signaling axis still holds much promise as a potential candidate for future cancer therapeutic interventions, especially for lung cancer. 

## 9. Conclusions

In this review, the significant role of the IGF/IGF-1R signaling axis in the development and progression of lung cancer, particularly in the context of tobacco smoke-induced carcinogenesis, is highlighted. Our findings emphasize the intricate mechanisms by which IGF-1R contributes to various aspects of lung cancer pathogenesis, including tumor proliferation, metastasis, EMT, stemness, and resistance to both chemotherapy and targeted therapies. Because of this, the IGF-1R signaling pathway serves as an attractive target for the development of lung cancer treatments. However, many challenges are associated with pursuing IGF-1R inhibitors, which have been highlighted by the disappointing results of recent clinical phase trials. 

The setbacks observed in recent clinical trials with IGF-1R inhibitors highlight the need to refine patient selection criteria and identify predictive biomarkers to assess the efficacy of IGF-1R inhibition in clinical trials. Additionally, the development of successful cancer therapies requires a thorough understanding of the structural components and downstream effector pathways of the IGF/IGF-R axis, all of which offer potential avenues for novel therapeutic targets. Although many studies have explored IGF-1R inhibition as a treatment strategy, this system’s complexity and drug design challenges highlight the need for more nuanced and tailored approaches. Furthermore, the diverse functions of IGFBPs, which may operate independently of IGF signaling and are modulated by factors like p53, can provide new opportunities for therapeutic interventions.

The activation of IGF-1R by carcinogens in tobacco smoke can have significant downstream effects on lung cancer development, promoting cell growth, proliferation, and survival. The cumulative effect of these processes creates favorable circumstances for tumorigenesis within the lungs, as well as for metastasis due to tumor progression. When aberrantly activated, this pathway is also associated with the emergence of cancer stem cells. This can lead to tumor heterogeneity, therapy resistance, and a greater likelihood of disease recurrence. Gaining further clarity on the pivotal role of IGF-1R in lung cancer development is instrumental in identifying effective treatment strategies for this malignant disease.

In addition to the impact of IGF-1R on the development and progression of lung cancer, there is compelling evidence indicating that the IGF/IGF-1R axis plays a significant role in cancer treatment resistance. Dysregulation within this signaling pathway has been implicated in resistance to traditional chemotherapeutic agents and targeted therapies such as EGFR inhibitors. This underscores the need to address IGF-1R signaling within lung cancer treatment. Targeting the IGF/IGF-1R axis is a promising therapeutic approach, particularly when combined with other targeted agents or chemotherapy regimens. Pre-clinical studies have shown significant benefits can be derived from such a combinatorial strategy, suggesting that overcoming treatment resistance requires multifaceted and comprehensive interventions to counteract the tumor-promoting properties of IGF-1R.

## 10. Future Directions 

Because of the structural homology between IGF-1R and IR, it is necessary to gain clarity on hybrid IGF-1R/IR receptors and their implications for disease progression and treatment response. Understanding the intricacies of these hybrid receptors is imperative for designing effective, targeted therapeutic interventions that minimize adverse side effects and metabolic disturbances. In addition, efforts should be directed towards identifying components of the IGF/IGF-1R axis that hold promise as predictive or prognostic biomarkers, thereby facilitating the development of personalized medicine tailored to individual patients. Setbacks observed in recent clinical trials exploring IGF-1R-targeted therapies indicate a need for more refined predictive biomarkers, such as EGFR or KRAS mutation status, to identify patients most likely to benefit from these therapies. Exploring the potential synergies between IGF-1R inhibition and established therapeutic modalities is also warranted, as these combinatorial approaches can overcome drug resistance and enhance treatment efficacy.

Furthermore, it is essential to delve into the IGF-independent functions of IGFBPs, which could reveal novel therapeutic targets with clinical implications beyond cancer treatment. Lastly, unraveling the complexities of IGR-1R signaling may reveal downstream targets beyond the receptor itself, which, if targeted, may present a powerful tool for circumventing resistance mechanisms and improving treatment strategies. Therefore, continued exploration of the intricate connections within the IGF/IGF-1R signaling axis is indispensable for realizing the full therapeutic potential of IGF-1R inhibition in the clinical setting, especially in cancer treatment.

## Figures and Tables

**Figure 1 biomedicines-12-00563-f001:**
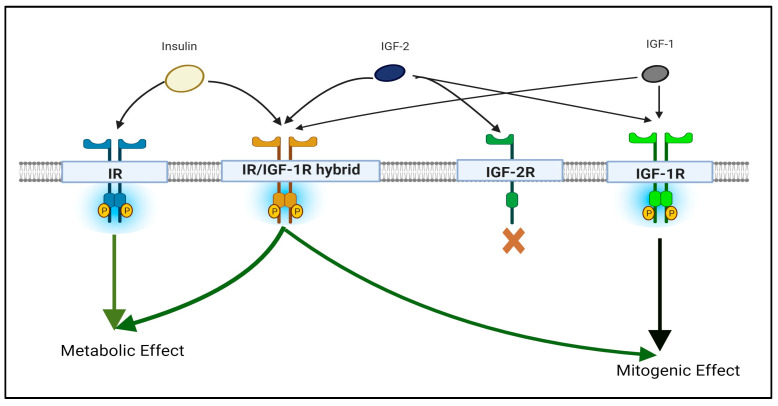
Schematic representation of the functional connections between the IR and IGF-R signaling axis. The binding of insulin-like growth factors 1/2 (IGF-1/2) to their receptor insulin-like growth factor receptor 1 (IGF-1R) initiates a signaling cascade that ultimately leads to cell growth and division. In contrast, insulin binds with the insulin receptor (IR) to regulate glucose metabolism and maintain blood sugar levels. IR dimers and IGF-1R dimers can combine to form hybrid receptors capable of binding with insulin and IGF-1/2. These hybrid receptors, known as IR/IGF-1R hybrid receptors, play a crucial role in regulating various physiological processes such as cell growth, metabolism, and survival. IGF-2R does not activate any downstream signaling pathways.

**Figure 2 biomedicines-12-00563-f002:**
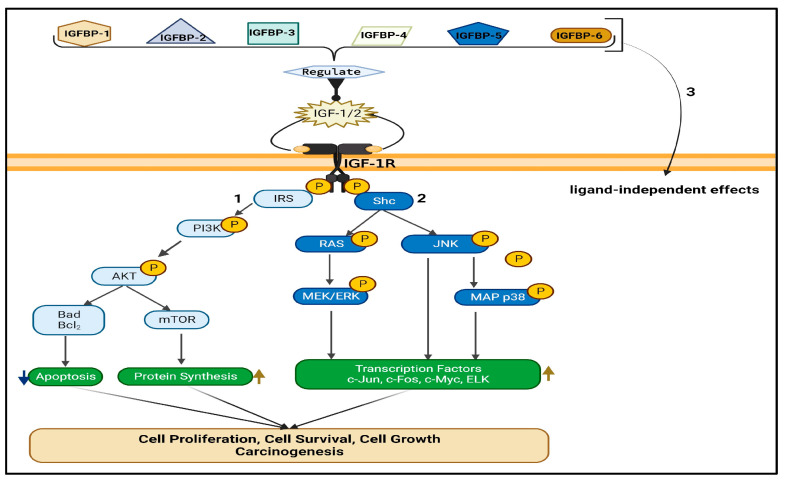
Schematic representation of the IGF/IGF-1R axis: (1) Insulin-like growth factor (IGF-1/2) binds to its receptor, insulin-like growth factor receptor 1 (IGF-1R), triggering a cascade of events that ultimately lead to the activation of the insulin receptor substrate (IRS). Once activated, IRS initiates the PI3K-AKT signaling pathway, which plays a crucial role in promoting cell survival. This is achieved through a reduction in apoptosis, a process of programmed cell death, and an increase in protein synthesis, which aids in cellular growth and maintenance. Overall, the activation of IRS and the subsequent PI3K-AKT signaling pathway are essential for maintaining healthy cellular functions. (2) The SHC protein acts as a signal transducer, triggering the activation of the Ras/MEK/ERK and JNK/MAPK pathways. These pathways, in turn, activate specific transcription factors that regulate the expression of target genes. By orchestrating this complex signaling cascade, the SHC protein plays a crucial role in coordinating cellular responses to diverse extracellular stimuli. This process is essential for maintaining cellular homeostasis and adapting to changing environmental conditions. (3) In addition to the activation of ligand-dependent signaling pathways, insulin-like growth factor-binding proteins (IGFBPs) 1–6 can also trigger ligand-independent signaling pathways. This means that IGFBPs can influence cellular processes even when not bound to IGFs. IGF-1R: Insulin-like growth factor receptor 1; PI3K: Phosphatidylinositol-3-kinase; AKT: Serine/threonine kinase, named protein kinase B (PKB); mTOR: Mammalian target of rapamycin; Bad: Bcl-2-associated death promoter; Bcl2: B-cell lymphoma 2; Shc: Adaptor protein; Ras: GTPase protein; JNK: c-Jun N-terminal kinase; MEK: Mitogen-activated protein kinase kinase; ERK: Extracellular regulated kinase; MAPK: Mitogen-activated protein kinase; ELK: ETS domain-containing protein.

**Figure 3 biomedicines-12-00563-f003:**
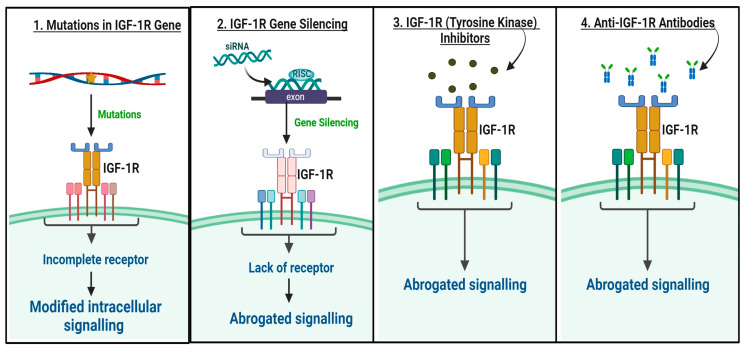
Schematic representation of various therapeutic approaches to targeting IGF-1R receptors. The figure demonstrates how mutations in IGF-1R can affect its structure and downstream signaling pathways. (1) Mutations in IGF-1R can result in an incomplete receptor, causing changes in downstream signaling pathways. These modifications can have a considerable impact on cellular responses. (2) The use of small interfering RNA (siRNA) to silence the gene responsible for encoding the Insulin-like Growth Factor 1 Receptor (IGF-1R) leads to a significant reduction in the receptor levels. This reduction in IGF-1R levels effectively disrupts the signaling pathways associated with the receptor, thereby abrogating the downstream signaling. (3) The application of tyrosine kinase (TK) inhibitors suppresses the activation of IGF-1R, effectively blocking downstream signaling pathways. Inhibiting IGF-1R with TK inhibitors disrupts the signaling cascade, thereby preventing the transmission of crucial cellular signals and, ultimately, halting the cellular response driven by IGF-1R. (4) Applying anti-IGF-1R antibodies inhibits IGF-1R, effectively disrupting its downstream signaling pathways. Anti-IGF-1R antibodies act by binding to IGF-1R and preventing its activation, leading to the abrogation of downstream signaling pathways (RISC: RNA-induced silencing complex).

**Table 1 biomedicines-12-00563-t001:** Pre-clinical studies of IGF-1R-targeted therapies in lung cancer models.

Drug Types	Compounds	Lung Cancer Models	Outcomes	References
**Monoclonal antibody**	Cixutumumab (IMC-A12)	SCLC cell lines	Increased sensitivity to chemotherapies, decreased cell growth	[120]
	R1507	NSCLC cell lines	The addition of R1507 to erlotinib inhibited cell growth and increased apoptosis	[121]
	Dalotuzumab (MK-0646)	NSCLC xenograft tumors in mice	Increased median survival of treated mice compared with controls, increased sensitivity to erlotinib	[122]
	Figitumumab (CP-751, 871)	NSCLC cell lines	Increased sensitivity to radiation therapy	[123]
		NSCLC xenograft tumors in mice	Addition of CP-751, 871 to radiation therapy delayed tumor growth in vivo	
**Tyrosine kinase inhibitor**	Linsitinib (OSI-906)	NSCLC cell lines	Inhibited IGF-1/IGF-2-mediated proliferation, increased apoptosis	[124]
		NSCLC xenograft tumors in mice	Inhibited tumor growth in cells expressing IGF-1R	[125]
	Picropodophyllin	B(a)P-induced lung tumors in mice	Decreased tumor multiplicity and load	[126]
		NSCLC cell lines	Decreased cell viability and in vitro invasive capacity	
	NT157	Lung cancer cell lines	Decreased cell viability and oncogene expression, creating a tumor-suppressive signaling network	[127]
	BMS-754807	NSCLC cell lines	Decreased cell survival, increased apoptosis, enhanced cytotoxic effects of platinum chemotherapies	[128]
		Lung cancer cell lines	The addition of BMS-754807 to dasatinib inhibited cell growth, and induced autophagy and cell cycle arrest	[129]
**Ligand neutralizing monoclonal antibody**	Xentuzumab (BI 836845)	NSCLC and SCLC cell lines	Decreased cell proliferation, enhanced anti-tumor efficacy of rapamycin	[130]
	Dusigitumab (MEDI-573)	Solid xenograft tumors in mice, including NSCLC	Inhibited IGF signaling pathways in tumors driven by autocrine IGF production	[131]
**shRNA-mediated Gene Silencing**	shIGF-1R (601, 801 and 3425)	NSCLC cell lines	Increased sensitivity to chemotherapies, decreased cell colony formation	[132]
**Antibody–drug conjugates**	W0101 (IGF-1R antibody-drug conjugate)	Lung cancer cell lines	Potent cytotoxic activity, inhibited tumor growth in cells expressing high levels of IGF-1R	[133]

Abbreviations: B(a)P—benzo(a)pyrene; IGF—insulin-like growth factor; IGF-1R—insulin-like growth factor receptor 1; NSCLC—non-small-cell lung cancer; SCLC—small-cell lung cancer; shRNA—short hairpin RNA.

**Table 2 biomedicines-12-00563-t002:** Clinical trials of IGF-1R-targeted therapies in patients with lung cancer.

Drug Types	Compounds	Phase	Lung Cancer Types	Outcomes	References
Monoclonal antibody	Cixutumumab (IMC-A12)	II	NSCLC	The addition of cixutumumab to other therapies increases toxicity without improving efficacy outcomes	[134,135]
	Ganitumab (AMG-479)	I	Advanced solid tumors	SD > 6 weeks in 7 patients, including 2 with NSCLC	[136]
	AVE1642	I	Advanced solid tumors	SD > 4 months in 11 patients, including 1 with NSCLC	[137]
	Teprotumumab (RV 001, R1507)	II	NSCLC	The addition of R1507 to erlotinib did not improve efficacy outcomes	[138]
		II	NSCLC	Terminated due to program termination (NCT00760929)	[22,138]
	Dalotuzumab (MK-0646)	I	SCLC	The addition of dalotuzumab to chemotherapies did not improve efficacy outcomes	[139]
		II	NSCLC	The addition of dalotuzumab to erlotinib did not improve efficacy outcomes	[140]
		II	Non-squamous lung cancer	The addition of dalotuzumab to chemotherapies did not improve efficacy outcomes	[141]
	Figitumumab (CP-751, 871)	III	NSCLC	The addition of figitumumab to chemotherapies did not improve efficacy outcomes	[119]
		III	NSCLC	Discontinued due to futile HR, lack of improved efficacy outcomes, and serious adverse events	[22,142,143]
	BIIB022	I	Advanced solid tumors	Preliminary evidence of biological activity in select patients; SD > 6 weeks in 20 patients	[144]
Tyrosine kinase inhibitor	Linsitinib (OSI-906)	II	SCLC	Linsitinib did not improve efficacy outcomes	[145]
			NSCLC	The addition of linsitinib to erlotinib did not improve efficacy outcomes	[146,147]
	Picropodophyllin (AXL 1717)	I	Advanced solid tumors	Median PFS of 31 weeks and OS of 60 weeks in 15 patients with NSCLC	[148]
Ligand neutralizing monoclonal antibody	Xentuzumab (BI 836845)	I	NSCLC	The addition of xentuzumab to afatinib did not substantially improve efficacy outcomes	[149]
	Dusigitumab (MEDI-573)	I	Advanced solid tumors	Preliminary evidence (from 43 patients, including 1 with NSCLC) warrants further clinical evaluation	[150]

Abbreviations: HR—hazard ratio; NSCLC—non-small cell lung cancer; OS—overall survival; PFS—progression-free survival; SCLC—small-cell lung cancer; SD—stable disease.

**Table 3 biomedicines-12-00563-t003:** Mechanisms of action for IGF-1R targeted therapies in anti-cancer treatment.

Molecule Type	Associated Target(s)	Examples	Outcomes	Intended Anti-Cancer Effects	References
Monoclonal antibody	Extracellular ligand-binding α-subunit domain of IGF-1R	Cixutumumab (IMC-A12), R1507, Dalotuzumab (MK-0646), Figitumumab (CP-751, 871)	Inhibits ligand binding to receptor and promotes receptor internalization and degradation	Anti-proliferative, anti-growth, anti-metastatic, prevention of EMT transition, pro-apoptotic, increased sensitization to chemotherapy	[121,155,156]
Tyrosine kinase inhibitor	ATP pocket of IGF-1R (ATP-competitive)	Linsitinib (OSI-906), BMS-754807, AG1024	Inhibits auto-phosphorylation of IGF-1R upon ligand binding, preventing recruitment of signaling proteins such as IRS and Shc	Anti-proliferative, anti-growth, anti-metastatic, pro-apoptotic, increased sensitization to chemotherapy	[127,157,158,159]
	Allosteric site on IGF-1R (non-ATP competitive)	XL228, Picropodophyllin, NT157		Decreased tumor multiplicity and load	[126]
Ligand-neutralizing monoclonal antibody	IGF-I/II	Xentuzumab (BI 836845), Dusigitumab (MEDI-573)	Prevents ligand binding to IGF-1R	Anti-proliferative, anti-growth, pro-apoptotic, increased sensitization to chemotherapy	[160,161,162]
shRNA-mediated gene silencing	IGF-1R mRNA	shIGF-1R(601, 801 and 3425)	Prevents translation of IGF-1R mRNA, silencing gene expression	Anti-tumorigenic, pro-apoptotic, increased sensitization to chemotherapy	[132]
Antibody–drug conjugates	Extracellular domain of IGF-1R	W0101 (IGF-1R antibody-drug conjugate)	Promotes internalization of receptor and conjugated cytotoxic drug in cells expressing high levels of IGF-1R	Anti-mitotic, anti-tumorigenic, cytotoxic	[133]

Abbreviations: EMT—epithelial–mesenchymal transition; IGF—insulin-like growth factor; IGF-1R—insulin-like growth factor receptor 1; IRS—insulin receptor substrate; shRNA—short hairpin RNA; Shc—Src homology and collagen protein.

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
