# Peer review of "Role of Insulin-like Growth Factor-1 Receptor in Tobacco Smoking-Associated Lung Cancer Development"

_biomedicines, 2024, doi:10.3390/biomedicines12030563_

Round 1

Reviewer 1 Report

Comments and Suggestions for Authors

The current review article “Unraveling the Nexus: Tobacco Smoking and Insulin-Like Growth Factor-1 Receptor in Lung Cancer Development” is interesting and demonstrated the role of IGF-1R as potent target for anticancer therapeutics in lung cancer via different mechanism. However, the manuscript is not properly explaining the current topic in terms of specific content, clinical trial, and successful studies in this field. Although the authors have summarized the all the possible aspects of IGF-1R in lung cancer carcinogenesis and their associated anticancer therapeutics. Some of my recommendations for possible incorporation are:

1.     First of all, the title of the manuscript is not properly designed, authors proposed a title which correlate the tobacco smoke induced IGF-1R signaling in lung cancer. However, major content in the manuscript does not explain the same. If authors go with the same title, they must include specific studies which clearly depicted the correlation between tobacco smoke and IGF-1R signaling in lung carcinogenesis otherwise they must change the title.

2.     If we remove the term “Unraveling the Nexus: Tobacco Smoking and Insulin-Like Growth Factor-1 Receptor” then what is novel thing in this review with available literature such as: https://doi.org/10.1016/j.ctrv.2011.07.008

https://doi.org/10.1016/j.molmed.2006.10.003

https://doi.org/10.1177/1758834011427576

3.     Authors are suggested to incorporate a table summarizing the anticancer mode of action of IGF-1R signaling via different mechanism like antiproliferative, apoptotic and antimetastatic potential and associated targets.

4.     Author’s perspective related to current topic is missing from this review.

5.     Authors are suggested to incorporate the recent clinical trials related to IGF-1R induced lung cancer.

6.     Authors are suggested to reformat the whole manuscript by deleting the irrelevant content and add only specific content of IGF-1R signaling targeting in lung cancer for each section. It is highly suggested to add recent studies of IGF-1R signaling and also incorporate recent references.

7.     In my opinion there is no requirement of figure 1 in this review. Authors may delete this. Or authors can combined figures like figure 1 and 2 which are designed for the same section, it will add more quality to review article.

8.     Rewrite the conclusion part as per the standard format. Author must have future projections for researchers working in this domain and how this review will enhance their knowledge.

9.     Abbreviations must be clearly stated in its expanded form.

Comments on the Quality of English Language

Moderate English corrections are needed.

Author Response

Dear Reviewer,

Thank you for your insightful review of our article. We appreciate your valuable feedback and suggestions for improving the manuscript. We carefully consider your recommendations to make the article more informative and comprehensive. Your comments are very helpful, and we hope that the revised manuscript will meet your expectations. Thank you again for your time and valuable feedback.

The current review article “Unraveling the Nexus: Tobacco Smoking and Insulin-Like Growth Factor-1 Receptor in Lung Cancer Development” is interesting and demonstrated the role of IGF-1R as potent target for anticancer therapeutics in lung cancer via different mechanism. However, the manuscript is not properly explaining the current topic in terms of specific content, clinical trial, and successful studies in this field. Although the authors have summarized the all the possible aspects of IGF-1R in lung cancer carcinogenesis and their associated anticancer therapeutics. Some of my recommendations for possible incorporation are:

  1. First of all, the title of the manuscript is not properly designed, authors proposed a title which correlate the tobacco smoke induced IGF-1R signaling in lung cancer. However, major content in the manuscript does not explain the same. If authors go with the same title, they must include specific studies which clearly depicted the correlation between tobacco smoke and IGF-1R signaling in lung carcinogenesis otherwise they must change the title.

Reply: Thank you for your valuable feedback. We highly appreciate your suggestion and have changed the manuscript's title to ensure it aligns better with its content. Your advice was invaluable in helping us revise the title, and we have updated it to describe our manuscript more accurately.

  1. If we remove the term “Unraveling the Nexus: Tobacco Smoking and Insulin-Like Growth Factor-1 Receptor” then what is novel thing in this review with available literature such as: https://doi.org/10.1016/j.ctrv.2011.07.008

https://doi.org/10.1016/j.molmed.2006.10.003

https://doi.org/10.1177/1758834011427576

Reply: Thank you for your comment. In our article, we attempted to connect the dots between tobacco smoke and IGF-1R research in lung cancer.  After thoroughly reviewing all publicly available review articles on lung cancer and IGF-1R, we could find few evidence to support this connection. We have also compiled a comprehensive overview of all pre-clinical and clinical studies on IGF-1R and lung cancer. This overview includes all the relevant information and findings related to the topic, which would benefit researchers and medical professionals in lung cancer research.

  1. Authors are suggested to incorporate a table summarizing the anticancer mode of action of IGF-1R signaling via different mechanism like antiproliferative, apoptotic and antimetastatic potential and associated targets.

Reply: Thank you for your suggestion. We have added a table to the manuscript according to your feedback.

  1. Author’s perspective related to current topic is missing from this review.

Reply: Thank you for your suggestion. I have made changes to the conclusion and included my own perspective.

  1. Authors are suggested to incorporate the recent clinical trials related to IGF-1R induced lung cancer.

Reply: According to the reviewer's comment, We have added recent clinical trials related to IGF-1R-induced lung cancer to our manuscript

  1. Authors are suggested to reformat the whole manuscript by deleting the irrelevant content and add only specific content of IGF-1R signaling targeting in lung cancer for each section. It is highly suggested to add recent studies of IGF-1R signaling and also incorporate recent references.

Reply: According to the reviewer's comment, I reformatted the manuscript to include recent studies and references on IGF-1R signaling in lung cancer and also removed irrelevant content.

  1. In my opinion there is no requirement of figure 1 in this review. Authors may delete this. Or authors can combined figures like figure 1 and 2 which are designed for the same section, it will add more quality to review article.

Reply: Thanks for the comments. Figure 1 illustrates the differences between IGF-1R, IGF-2R, and IR. Figure 2 provides a detailed description of the IGF-1R signaling pathway. I think both figures are important in understanding the IGF signaling pathway.

  1. Rewrite the conclusion part as per the standard format. Author must have future projections for researchers working in this domain and how this review will enhance their knowledge.

Reply: Based on the reviewer's comment, I have updated the conclusion and added the future perspective.

  1. Abbreviations must be clearly stated in its expanded form.

Reply: According to the reviewer's comment, all the Abbreviations are clearly stated and expanded accordingly.

Comments on the Quality of English Language: Moderate English corrections are needed.

Reply: We have carefully reviewed the English language in the manuscript and made corrections where necessary to ensure proper grammar, spelling, and punctuation.

Reviewer 2 Report

Comments and Suggestions for Authors

The review is very interesting, however is not updated on the bibliography. 

The authors delve into the significant role of the IGF/IGF -1R signaling axis in the development and progression of lung cancer, particularly in the context of tobacco smoke-induced carcinogenesis. I consider the evaluation of clinical trials on IGF-1R targeted therapies important and well done. The paper improves the schematization of the outcome problem. Compared with other published material, the paper improves data collection, schematization and simplification of the biochemical pathway and therapeutic problem. The data collection methodology is correct. The conclusions are correct but they are not updated and complete, which is why I have indicated that the problems should be explored further by following the bibliography that I have added. References are appropriate but insufficient.  The tables and figures and quality of the data are fine.

In particular, please consider these references:

1) Lines 108-109: Randhawa R, Cohen P. The role of the insulin-like growth factor system in prenatal growth. Mol Genet Metab. 2005 Sep-Oct;86(1-2):84-90. doi: 10.1016/j.ymgme.2005.07.028. PMID: 16165387.

2) Line 184: Zhang C, Cui T, Cai R, Wangpaichitr M, Mirsaeidi M, Schally AV, Jackson RM. Growth Hormone-Releasing Hormone in Lung Physiology and Pulmonary Disease. Cells. 2020 Oct 21;9(10):2331. doi: 10.3390/cells9102331. PMID: 33096674; PMCID: PMC7589146.

3) Line 213: Xu X, Qiu Y, Chen S, Wang S, Yang R, Liu B, Li Y, Deng J, Su Y, Lin Z, Gu J, Li S, Huang L, Zhou Y. Different Roles of the Insulin-like Growth Factor (IGF) Axis in Non-small Cell Lung Cancer. Curr Pharm Des. 2022;28(25):2052-2064. doi: 10.2174/1381612828666220608122934. PMID: 36062855.

4) Line 259: Cevenini A, Orrù S, Mancini A, Alfieri A, Buono P, Imperlini E. Molecular Signatures of the Insulin-like Growth Factor 1-mediated Epithelial-Mesenchymal Transition in Breast, Lung and Gastric Cancers. Int J Mol Sci. 2018 Aug 15;19(8):2411. doi: 10.3390/ijms19082411. PMID: 30111747; PMCID: PMC6122069.

5) Line 279: Nwabo Kamdje AH, Seke Etet PF, Kipanyula MJ, Vecchio L, Tagne Simo R, Njamnshi AK, Lukong KE, Mimche PN. Insulin-like growth factor-1 signaling in the tumor microenvironment: Carcinogenesis, cancer drug resistance, and therapeutic potential. Front Endocrinol (Lausanne). 2022 Aug 9;13:927390. doi: 10.3389/fendo.2022.927390. PMID: 36017326; PMCID: PMC9395641.

6) Discussion for figure 3: Chakraborty C, Doss CG, Bandyopadhyay S, Agoramoorthy G. Influence of miRNA in insulin signaling pathway and insulin resistance: micro-molecules with a major role in type-2 diabetes. Wiley Interdiscip Rev RNA. 2014 Sep-Oct;5(5):697-712. doi: 10.1002/wrna.1240. Epub 2014 Jun 18. PMID: 24944010.

Author Response

Dear Reviewer,

Thank you for taking the time to review our manuscript, and we appreciate your feedback. We are glad to hear that you found our data collection methodology to be correct and that the tables and figures are of good quality. Your comments are very helpful, and we hope that the revised manuscript will meet your expectations.

The authors delve into the significant role of the IGF/IGF -1R signaling axis in the development and progression of lung cancer, particularly in the context of tobacco smoke-induced carcinogenesis. I consider the evaluation of clinical trials on IGF-1R targeted therapies important and well done. The paper improves the schematization of the outcome problem. Compared with other published material, the paper improves data collection, schematization and simplification of the biochemical pathway and therapeutic problem. The data collection methodology is correct. The conclusions are correct but they are not updated and complete, which is why I have indicated that the problems should be explored further by following the bibliography that I have added. References are appropriate but insufficient.  The tables and figures and quality of the data are fine.

In particular, please consider these references:

1) Lines 108-109: Randhawa R, Cohen P. The role of the insulin-like growth factor system in prenatal growth. Mol Genet Metab. 2005 Sep-Oct;86(1-2):84-90. doi: 10.1016/j.ymgme.2005.07.028. PMID: 16165387.

2) Line 184: Zhang C, Cui T, Cai R, Wangpaichitr M, Mirsaeidi M, Schally AV, Jackson RM. Growth Hormone-Releasing Hormone in Lung Physiology and Pulmonary Disease. Cells. 2020 Oct 21;9(10):2331. doi: 10.3390/cells9102331. PMID: 33096674; PMCID: PMC7589146.

3) Line 213: Xu X, Qiu Y, Chen S, Wang S, Yang R, Liu B, Li Y, Deng J, Su Y, Lin Z, Gu J, Li S, Huang L, Zhou Y. Different Roles of the Insulin-like Growth Factor (IGF) Axis in Non-small Cell Lung Cancer. Curr Pharm Des. 2022;28(25):2052-2064. doi: 10.2174/1381612828666220608122934. PMID: 36062855.

4) Line 259: Cevenini A, Orrù S, Mancini A, Alfieri A, Buono P, Imperlini E. Molecular Signatures of the Insulin-like Growth Factor 1-mediated Epithelial-Mesenchymal Transition in Breast, Lung and Gastric Cancers. Int J Mol Sci. 2018 Aug 15;19(8):2411. doi: 10.3390/ijms19082411. PMID: 30111747; PMCID: PMC6122069.

5) Line 279: Nwabo Kamdje AH, Seke Etet PF, Kipanyula MJ, Vecchio L, Tagne Simo R, Njamnshi AK, Lukong KE, Mimche PN. Insulin-like growth factor-1 signaling in the tumor microenvironment: Carcinogenesis, cancer drug resistance, and therapeutic potential. Front Endocrinol (Lausanne). 2022 Aug 9;13:927390. doi: 10.3389/fendo.2022.927390. PMID: 36017326; PMCID: PMC9395641.

6) Discussion for figure 3: Chakraborty C, Doss CG, Bandyopadhyay S, Agoramoorthy G. Influence of miRNA in insulin signaling pathway and insulin resistance: micro-molecules with a major role in type-2 diabetes. Wiley Interdiscip Rev RNA. 2014 Sep-Oct;5(5):697-712. doi: 10.1002/wrna.1240. Epub 2014 Jun 18. PMID: 24944010.

 Reply: We acknowledge that the conclusions were not updated and complete. To address this issue, we included the bibliography that you provided. We also have incorporated your suggestions into the manuscript and modified it accordingly.

Round 2

Reviewer 1 Report

Comments and Suggestions for Authors

Authors have substantially revised the manuscript as per the reviewer's comments. This it can now be considered for publication.

Comments on the Quality of English Language

Minor English changes are required